# Factors Associated with Injury Rate and Pregnancy Success in Rhesus Macaques

**DOI:** 10.3390/biology11070979

**Published:** 2022-06-28

**Authors:** David A. Massey, Faye Peters, Jim Willshire, Claire L. Witham

**Affiliations:** 1Centre for Macaques, Harwell Institute, Medical Research Council, Salisbury SP4 0JQ, UK; d.massey2@newcastle.ac.uk (D.A.M.); f.peters@har.mrc.ac.uk (F.P.); 2Biosciences Institute, Newcastle University, Newcastle upon Tyne NE2 4HH, UK; 3Endell Farm Vets, Salisbury SP1 3UH, UK; jim@endellfarmvets.co.uk

**Keywords:** rhesus macaque, colony management, aggression

## Abstract

**Simple Summary:**

Rhesus macaques are bred around the world for use in biomedical research. They have a strict dominance hierarchy which they reinforce with aggression, and fight injuries are a major welfare concern in group-housed macaques. In this study we examined 10 years of injury records at a colony in which the monkeys were housed in small breeding groups with a single adult male. We found that breeding females were disproportionally likely to be injured. We examined how factors, such as group size, might affect both the injury rate and the probability of the females becoming pregnant. We found that introducing a new adult male to a group had the greatest effect on the injury rate but also significantly increased the probability of the females getting pregnant.

**Abstract:**

Fight injuries are a major welfare concern in group-housed rhesus macaques. This is particularly a problem in breeding groups. We investigated which factors might affect the injury rate in group-housed macaques and also looked at how the same factors might affect productivity. We analysed 10 years of health records at a breeding colony in which monkeys were kept in small breeding groups consisting of a single adult male and 2–13 females and their offspring or single-sex juvenile groups. We found that females over the age of 2.5 years in breeding groups were the most likely to be injured. We focused on these females and used generalised mixed-effect models to examine which factors affected the injury rate and their productivity (probability of getting pregnant). The biggest risk factor for injury was the introduction of a new adult male to a breeding group. However, this also produced a large increase in the proportion of females that became pregnant, suggesting that there may be a trade-off between the risk of injury and the productivity. We also found that females in large groups with a young breeding male had a very high risk of injury. We recommend keeping young breeding males (<7 years) in smaller groups.

## 1. Introduction

Macaque groups have complex social relationships, which makes managing them in captivity a challenge. There are an estimated 100,000 primates used in research worldwide each year, with the majority of these primates being either cynomolgus macaques (*Macaca fascicularis*) or rhesus macaques (*Macaca mulatta*).

One of the major challenges in breeding macaques for research is that a high number of injuries result from aggression [1]. This is a significant welfare and management problem in breeding colonies [2,3]. To understand why aggression is particularly an issue in rhesus macaques, we need to look at how they interact with each other and how they are bred in captivity. Rhesus macaques have a strict dominance hierarchy and aggression plays a major role in maintaining this hierarchy, much more so than in than other macaque species [1,4,5,6,7,8]. This challenge has been reviewed in depth by McCowan and colleagues [3]. 

In many institutions in which macaques are used in research, they are housed in pairs, which avoids the more complex relationships seen in larger groups. A number of studies have been undertaken to investigate what makes a successful pairing [9,10]. In rhesus macaque breeding colonies pair-housing is rarely an option, with typical breeding groups ranging from a single male with a small number of females (<10 animals per group) to large multi-male–multi-female female groups (>100 animals per group). Wild rhesus macaques live in multi-male–multi-female groups with a matrilineal organisation. Female offspring remain in their natal group, whereas male offspring emigrate from their natal group at approximately 3–4 years of age [11]. The composition of breeding groups in captivity is driven by the size, purpose and philosophy of the colony and can have several important differences when compared to groups in the wild:(1)The group size can be much smaller (for example, single-male groups with a small number of females) or much larger (such as the large multi-male–multi-female groups at the Californian National Primate Center in the USA).(2)Some colonies keep macaques in matrilineal groups to mimic the situation in the wild [12], whilst others do not retain female offspring in their natal groups.(3)The age at which male offspring are removed from their natal groups can be much earlier [13] or later than in the case of male emigration in the wild [14].

McCowan and colleagues from the Californian National Primate Research Center have looked in depth at the factors that contribute to aggression, group instability and matrilineal overthrow in large group of macaques [3,14,15,16]. As part of these studies, Beisner et al. [15] found that deleterious aggression is associated with three main factors: the presence of older natal males in a group, genetic fragmentation of the matrilines and the power structure within the group [13]. At the other end of the group size spectrum, single-male groups with a small number of females formed the basis of many of the early studies by Robert Hinde and colleagues at the Madingley colony in Cambridge, UK. Some of the studies at this centre looked at productivity and birth characteristics but this group did not focus on aggressive interactions [17,18].

Keeping macaques in matrilineal groups may provide multiple benefits. Related females may be less likely to fight; for example, Stavisky and colleagues found that injury rates were highest for those groups with the highest number of unrelated animals [2]. The second potential benefit is that young females that are retained in their breeding groups have opportunities to learn infant handling by babysitting younger siblings. Anecdotal evidence suggests that this may improve their chances of rearing their own first offspring successfully but there is a lack of studies in this area.

Matrilineal groups present their own challenges though, as they can require the introduction of new adult males into the group to prevent inbreeding (particularly for smaller groups with single males). Introducing one or more adult males to a group can be a fraught experience with a significant risk of failure and multiple injuries. Studies on several colonies have been performed to investigate the factors that contribute to successful introductions [12,19,20,21,22,23].

A lesser-studied aspect of the management of macaque colonies is that of the factors that affect productivity (the probability of a female becoming pregnant and/or successfully raising the infant). Many of the factors that affect injury rates in macaques are also likely to affect productivity (for example, we might expect a boost in productivity if a new adult male in his prime is added to the group). However, there may also be a trade-off between the injury rate and the productivity. Gagliardi and colleagues [24] investigated the factors affecting reproductive success (the proportion of pregnancies that resulted in a live birth and then the number of infants that survived 30 days and 1 year). They found that maternal age played an important role with primiparous (first time) mothers and that older mothers were significantly less likely to have an infant survive to 30 days. Bercovitch [25] reviewed the factors that contribute to reproductive success in macaques and showed that feeding patterns that affect body condition had a significant effect on both male and female reproductive success.

The aim of our study was to investigate the factors associated with injury rates and productivity in small single-male breeding groups. This is currently of particular relevance as the COVID-19 pandemic has led to a halt in the export of macaques from China, one of the major exporters, and to an interest in setting up new breeding colonies. We were particularly interested in how colony management decisions, such as group size, composition and the movement of males, might affect injury rate and productivity.

We analysed ten years of colony records from the UK Medical Research Council’s Centre for Macaques to identify the factors that predict injury rate and/or productivity. We chose to focus on the period between 2008 and 2017, as during this period the format of the health records was consistent and the animals were under the care of the same named veterinary surgeon. We began by looking at which classes of macaques in the colony were most likely to be injured and which body parts were most likely to be injured. On finding that 80% of the injuries in the colony occurred to breeding females we then focused on the breeding females and looked at which factors predicted injury rate and productivity.

## 2. Materials and Methods

### 2.1. Animals

The subjects of this study were 708 rhesus macaques (*Macaca mulatta*; Indian origin; 404 females) housed at the Medical Research Council Centre for Macaques (CFM) between 1 January 2008 and 31 December 2017. The monkeys were housed in groups of between 2 and 27 animals and were of two main types:Breeding groups (consisting of one adult male, multiple females and their offspring)Single-sex juvenile groups

The average age of weaning (removal from natal group) of the juveniles was 21.5 ± 9.7 months (mean ± standard deviation) [13].

The main enclosures at the CFM consist of two separate areas; the first has dimensions 8.04 m length × 3.35 m width × 2.8 m height and the second has the dimensions of 6.12 m length × 1.5 m width × 2.8 m height). Animals had free access between the two areas by the means of four hatches (at various heights) unless separated for cleaning or veterinary treatment. Both rooms were lit by artificial lighting (12 h light: 12 h dark cycle) and the main enclosure also has natural light via a large bay window. The main enclosure has multiple shelves at different heights, visual barriers and other enrichment, such as swings, horizontal and vertical wooden poles and barrels. A small number of groups were housed in smaller enclosures (half the size of the standard enclosure) during this period.

Each day the monkeys were provided with primate diet in small pellet form, a fine forage mixture and fruit and vegetables. All the food was scattered in the deep bedding litter in the main enclosure to encourage natural foraging behaviours. Water was available ad libitum. The colony underwent annual health screening in accordance with the FELASA guidelines on screening of non-human primates [26]. Figure 1 shows an example of the typical indoor enclosure.

### 2.2. Data Collection

Health and husbandry information was recorded by care staff on a daily basis on paper records. Health and breeding information was recorded on individual health records for each monkey and husbandry information (including feeding and cleaning) was recorded in day books for each group. These data were transferred to an electronic Oracle-based database (ENOS; https://poweredbyenos.com/; [27]) by technical and scientific staff. The database information was checked for errors by a senior member of staff (CW). Prior to analysis, the data were exported from the database to an excel spreadsheet and subsequently analysed in Matlab (www.mathworks.com; version 2020a; [28]) and R (https://www.r-project.org/; version 4.2.0; [29]).

### 2.3. Injury Definition

Injuries were included in the dataset if they required some form of veterinary attention (such as antibiotic and/or analgesic injection or suturing). All injuries listed in the database were sorted into those most likely to be caused by fighting (for example, slice wounds, puncture wounds, lacerations, amputations and de-gloving injuries) and those for which the cause was unclear and that were more likely to be accidental (for example, bruising, swelling, limping with no obvious wounds and broken bones). This second category accounted for 6.6% of all the injuries recorded and these were removed from the dataset. It was not possible to quantify the severity of the injury based on the information recorded during this period. 

### 2.4. Measures

To reduce the dataset to a meaningful size for modelling we used a time resolution of one month, calculating the number of injuries in the month and the probability of an animal becoming pregnant in a given month. For all the continuous predictor variables, such as age, we calculated their average value across the month. Table 1 shows the definition of all the predictor variables. Predictor variables were either individual level (applied to the individual animal, such as age) or group level (such as the number of animals in the group). These data are provided as a csv file in the Appendix A. Measures were only included if they were reliably recorded throughout the 10 years of the study (we excluded dominance rank as a measure from the main study as it was only reliably recorded from 2014 onwards).

The number of injuries per month followed a negative binomial distribution with an average over-dispersion factor of 1.5 (this over-dispersion meant that the Poisson distribution was not suitable). To improve the clarity of the figures in this article we have shown the injury rate as the number of injuries per animal per year. This was calculated for each category by averaging the number of injuries per month across the number of animals in that category and multiplying by 12 to obtain the rate per year.

The productivity was defined as the probability of a female becoming pregnant in a given month. This required estimation of the conception date of all babies born within the colony in this period (including still-births). For this, the average gestation period in rhesus macaques (166 days; [30]) was subtracted from the date of birth to give the approximate date (and month) of conception. This is an approximate measure, and we were unable to include any conceptions that resulted in early miscarriages.

### 2.5. Defining Groups and Age Categories

As the study covered a period of 10 years, the composition of the groups changed over time. We assigned each group a unique group ID. If a group had a substantial change we assigned a new ID and if the group was divided in two we gave each of the smaller groups a new ID.

We divided the age range into five categories: infants, juveniles, adolescents, young adults and adults. Infants were defined as aged between 0 and 1 years old, an age at which they are dependent on their mother and before their mother has another baby. Juveniles were defined as aged between 1 and 2.5 years old. At this age they become more independent of their mother (if still in natal group) and they may have one or more younger siblings but they have not entered adolescence. Adolescents were defined as aged between 2.5 and 4 years. This is an age range over which females can become pregnant (the youngest animal ever to become pregnant in the colony was 2.5 years old), both sexes start exhibiting more sexual behaviours and males start developing their large canine teeth. Young adults are defined as aged between 4 and 7 years old. During this period both sexes may still be growing (especially the males) and the canine teeth in the males are fully grown. All older monkeys (7 years and older) were classified as adults. Table 2 shows the number of individual animals and the total number of observations (each month of data per animal being a separate observation) for each sex and age class.

### 2.6. Statistics

We used generalised linear mixed-effect models to calculate which factors predicted the injury rate and productivity (only breeding females were used for the productivity model). The glmmTMB package in R (https://glmmtmb.github.io/glmmTMB/; version 1.1.3; [31]) was used to fit the models to the data. For the models, all of the numerical predictor variables were centred on zero by taking the values and subtracting the mean. We tested for collinearity between the predictor variables by calculating the variance inflation factors for each predictor variable [32]. All predictor variables included in the model had a variance inflation factor of less than two. For all models, individual ID and unique group ID were modelled as random intercepts. The random effects were checked by visual inspection to confirm that they followed an approximately normal distribution (Appendix A). All R scripts and the data are available in the Appendix A; Appendix A).

## 3. Results

The health records of 708 monkeys (404 females) housed at the Medical Research Council Centre for Macaques (CFM) between 1 January 2008 and 31 December 2017 were analysed. Over this period there were a total of 1920 injuries.

### 3.1. Distribution of Injuries by Age, Sex and Group Type

We first looked at the injury rate across ages, sexes and group types (breeding versus non-breeding). The results are shown in Figure 2. The injury rate used throughout the study is the number of injuries normalised by the number of animals and scaled to give the number of injuries per animal per year. The distribution of ages and sexes in the dataset is shown in Figure 2A. The number of males falls off rapidly after five years due to the structure of the macaque breeding groups, in which one male is housed with multiple females. The number of females gradually declines, with very few females over the age of 18 retained in the colony.

Figure 2B shows the injury rate for the different ages and sexes. It is clear from this figure that the adult females were the most likely animals in the colony to receive injuries and the older adult males and infants of both sexes the least likely. The increase in injuries in females appears to start between 2 and 3 years of age, which corresponds to the age at which the females first become sexually mature (the youngest recorded conception in the colony was at ~2.5 years with the infant born just before the female’s third birthday). There was a sharp peak in the number of injuries the males receive between 4 and 6 years. The reason for this is likely to be two-fold: firstly, some males in this age range are in their first season as breeding males and because they are not fully grown, they are more likely to be injured by the females in the group than older males are; secondly, males in this age range in single-sex groups start to fight amongst themselves.

To analyse this in more detail we show the number of injuries by sex, age category and group type. For this part of the analysis we grouped the ages together into five categories: infant (0–1 years), juvenile (1–2.5 years), adolescent (2.5–4 years), young adult (4–7 years) and adult (7 years and over). Table 2 summarises the number of observations for each sex and age class. The means and standard errors of the injury rate are shown in Figure 2C. Injuries to females of adolescent age (2.5 years) and older housed in breeding groups accounted for 80% of the total injuries observed.

We fitted a negative binomial mixed-effects model (see Section 2.6 for details) to the injury count data. We used sex, age class, group size and group type (breeding/non-breeding) as the fixed effects with an interaction term for sex and age class and random terms for individual identity and unique group identity (Table 3). As the interaction effect in Table 3 is difficult to visualise, we have presented a simplified version of the estimates in Table 4. There was a significant interaction between sex and age class, with adult females and young adult females being the most likely to be injured; infants (both sexes) and adult males the least (Table 4). Animals in breeding groups were more likely to be injured than animals in other group types. Surprisingly, animals in larger groups were less likely to be injured than animals in smaller groups. This may be due to larger groups having a larger number of infants and juveniles.

### 3.2. Location of Injury

Using the information on the database it was possible to classify the majority (94%) of injuries by the body part they affected. Of the other injuries, most (5%) were classified as multiple injuries (more than one body part affected) and the rest (1%) were unspecified. The numbers on the illustration in Figure 3A show the percentage of wounds that occurred to each body part. From this figure it is obvious that the legs (21%) are the most likely area to be targeted, followed by the arms (14%) and the tail (10%).

The locations were divided up into distal body parts (ears, hands, feet and tail), proximal (legs, arms, head) and core (back, chest, side, abdomen and rump) as indicated by the shading in Figure 3A. These were then compared across five different age categories used in Figure 2C (infant, juvenile, adolescent, young adult and adult). Adults were significantly more likely to have injuries to the core body areas than younger animals, whereas the younger animals were more likely to have injuries to distal body areas (Figure 3B; chi-square test, *p* < 0.001).

### 3.3. Variation in Injury Rate within a Single Group

To illustrate how injury rate can vary within a single group across a long period we looked in detail at one group that went through a number of changes between 2010 and 2017 (Figure 4). The group was formed of eleven young females in 2010. The females were all half-sisters (sharing the same father) with three pairs of full sisters and were aged between 12 months and 36 months. For the first event a 36-month-old male (Ruger) was introduced to the group (point P, Figure 4A). The first infant was born in 2011, with another three infants in 2012 and seven infants in 2013. As the monkeys aged the injury rate increased until it reached a maximum of 3.78 injuries per animal per year in the 3rd quarter of 2013. The majority of the injuries were inflicted by the male Ruger and the decision was taken by the colony management and veterinary staff to remove Ruger from breeding (point Q, Figure 4A). He was replaced by a much older and experienced male, Dan (point R, Figure 4A). Subsequently the injury rate dropped to less than 0.5 injuries per animal per year and remained low whilst Dan remained in the group.

In 2015 the colony management team decided to retain four female offspring so that the group would become matrilineal. As part of the decision the group was split in two, and five adult females and two juvenile females were retained in each group (point S, Figure 4). Full-sister and mother-infant pairs were kept together, and the decision of which females was to be placed in each group was based on behavioural observations of the social relationships within the group. The first group, Serena’s group (Figure 4C), remained with Dan in their original enclosure and the second group, Saphy’s group (Figure 4B; same scale), were moved to a new enclosure and introduced to a young male, Utah. For the first 18 months there were a number of minor injuries in this group (mainly ear and facial injuries, most likely inflicted by the high-ranked females rather than the male) and then the injury rate fell to below 0.5 injuries per year. In 2016, Dan was removed from Serena’s group due to old age (point T, Figure 4C) and was replaced by a younger male, Viktor (point U, Figure 4C). The injury rate increased and remained high until the end of the period covered by this analysis. 

### 3.4. Injury Rate in Breeding Females

For this part of the analysis we only looked at females of adolescent age and older within breeding groups. These females were significantly more likely to suffer injuries than any other animals in the colony, accounting for 80.0% of all the injuries, despite making up only 28.8% of the animals in the dataset and 40.3% of the individual data points. By focusing on these females, we were able to examine the effects of breeding-group-specific events (such as the introduction of a new adult male) on both the injury rate and the productivity. This dataset consisted of 10,779 entries from 224 females.

The integration of a new male to a group followed either the removal of the old male or the establishment of a new group. This integration had two main phases. During the initial introduction phrase, the male was gradually introduced to the females and may have been either separated from the females overnight or been kept with some of the females whilst the others were in a separate area. The next phase was the integration period, in which the male was left in with all the females overnight. For this analysis we have defined the integration period as the 3 months following the first night the male was left in overnight with all the females.

We fitted a negative binomial mixed-effects model (see Section 2.6 for details) to the injury count data for the breeding females. We used the following factors as the fixed effects:Individual age;Whether the individual was pregnant (true/false);Whether the individual had an infant under 12 months of age (true/false);Age of the oldest female in the group;Age of the breeding male;Number of breeding females in the group;Whether the breeding females are unrelated (true/false);Whether a new breeding male had been introduced to the group (established/introduction/integration);The season (with December–February being the main mating season and June–August the main birth season).

We also included an interaction term for the age of the breeding male and the number of breeding females and random terms for individual identity and unique group identity (Table 5).

Figure 5 (top row) and Figure 6 show the average injury rates for a selection of the factors used in the model. Introducing a new male into a group of females had the largest effect on the injury rate, with the largest increase seen in the introduction phrase. There was a small reduction in injury rate during the first 3 months after the males were fully integrated into the group, but this is still higher than the established baseline condition (Figure 5A). There was a significant effect of age, with older females more likely to be injured than younger females. Pregnancy, the presence of an infant (Figure 5C), the age of the oldest female and the presence of unrelated females had no effect on the injury rate (Figure 5B). Surprisingly, season did not have a significant effect on the injury rate either. There is a slight trend towards lower injuries in the main birth season (June–August), but it is not significant.

We modelled an interaction between the number of females in a group and the age of the breeding male, as the colony managers had the perception that older males were better at coping with larger groups than younger males (or, more likely, there is selection pressure within the colony, such that only the older males with a proven record are allowed to remain in breeding). Overall, there was a small reduction in individual injury rate as the number of females in the group increased and a reduction with increasing male age. However, there was also a significant interaction between the number of females and the age of the male.

Figure 6A compares the injury rate for the youngest males (3–7 years) and oldest males (15–22 years) across three different group sizes (2–5 females, 6–9 females and 10–13 females). In the case of the young males there was an increase in the individual injury rate with group size (in larger groups each female was more likely to be injured than in smaller groups). The trend for older males was the opposite and showed a decrease in injury rate with the increase in group size (females in larger groups were less likely to be injured than in smaller groups). The difference was particularly noticeable when we compared the average number of injuries for the group as a whole rather than at the individual level (Figure 6B). In large groups with young males the injury rate was five times higher than in large groups with an old male. As mentioned above, this probably reflects a selection bias in the retention of older males for breeding, and this will be considered in more detail in the Discussion Section.

Dominance rank was not included in the main study as we only had rank data for two thirds of the animals (and in many cases these were estimates of high, mid and low ranked animals). We have included a model in the Appendix A that includes rank (each monkey categorized as high, mid or low ranking), but there was no improvement to the model fit by including rank (assessed by Akaike’s information criterion) and no significant effect of rank on the injury rate (there was a slight tendency for low ranked animals to have more injuries than high ranked animals).

### 3.5. Productivity of Breeding Females

For model breeding success we looked at whether a given animal became pregnant in a given month. The month of conception was estimated from the date of birth minus the average gestation period of a rhesus macaque. This is an approximate method and does not include pregnancies that resulted in early miscarriages. As animals that are already pregnant cannot become pregnant again, we excluded pregnant females from this analysis. Therefore, the breeding success dataset is a subset of the main breeding female dataset (8926 entries from 223 females).

We used the same factors that were used for modelling injury rate, as detailed in Section 3.4 (without the fixed factor for pregnancy) and used a binomial model for the output with 1 for conceived and 0 for did not conceive. The results of the model are shown in Table 6 and a subset of the factors are illustrated in Figure 5 (lower row). Rhesus macaques are seasonal breeders and there is a significant effect of season on the probability of conception. As expected, the introduction of a new male into a group led to a significant increase in the proportion of females that conceived, particularly during the first 3 months after integration (Figure 5D). Females from a group in which all the females were related were slightly more likely to conceive than were those from groups with unrelated females, but this was not statistically significant (Figure 5E). Females with infants under the age of 12 months were also more likely to conceive, but again this was not significant (Figure 5F). Surprisingly, the age of the oldest female in a group had a negative association with the probability of females in that group becoming pregnant, but the age of the individual female did not have an effect. The age of the breeding male and the number of females in the group had no significant effects on the rate of conception.

### 3.6. Dental Treatment

Anecdotal evidence in the colony suggested that breeding males show increased aggression when they have dental problems. Over the 10 years of records that were examined we found 18 instances of breeding males having had dental treatment (predominantly extraction of broken teeth and canine root canal treatment). Of these, two were excluded from subsequent analysis due to changes in group composition. For the remaining 16 we calculated the average injury rate per individual for the three months preceding dental treatment and the three months following. Figure 7A shows the boxplots of the results for before and after treatment. Following dental treatment, the injury rate decreased by an average of 0.31 ± 0.25 injuries per individual per year (median ± standard error of median). The individual changes are shown in Figure 7B. Of the sixteen cases, nine showed a clear decrease in injury rate following treatment, two showed an increase and the remaining five showed no change (including two in which the injury rate was zero both before and after treatment). There is a clear relationship between the pre-treatment injury rate and the size of the decrease. The change in injury rate was significantly less than zero (two-sided non-parametric Wilcoxon signed rank test, *p* < 0.05).

## 4. Discussion

The clearest finding from this study is that breeding females are at the highest risk of injury (Figure 2). This was expected, but the magnitude of the risk was surprising, with 80% of all the injuries having been observed in this group despite the fact that they contributed to only 40% of the observations. Many of the injuries to the females were likely to be inflicted by the adult breeding male (such as slice wounds and puncture wounds inflicted by male canine teeth), but it was not possible to assess this with the information in the database. The case study presented in Section 3.3/Figure 4 and the effect of male dental treatment on injury rate also support the premise that many of the injuries were caused by the males. This does not necessarily mean that the males instigated the aggression. It is not uncommon in macaque colonies for the males to intervene when female–female interactions become aggressive and, depending on the male, this may result in injury [33]. For example, Flack and colleagues found that male pig-tailed macaques play a peace-keeping role in their groups [34]. The low level of injuries in adult males was expected due to the presence of only one adult male in each group. Male-on-male aggression is a major contributor to many of the male injuries in multi-male captive breeding groups and in the wild [14].

The distribution of the wounds (Figure 3) showed that the injuries were most likely to occur on the legs. Together, the legs, tail, rump and feet accounted for 43% of injuries, which suggests that a lot of injuries occur during chases. There have not been many studies that looked at the distribution of injuries. Freeman and colleagues focused on the distribution of wounds caused by self-injurious behaviours [35] and Springer and colleagues focused on the tetanus risk of different wound locations [36].

### 4.1. Selection Pressures within the Colony

One of the significant findings was that of the interaction effect between the age of the breeding males and the number of breeding females, with a particularly striking difference between the injury rates in large groups containing young males (<7 years age) and those in large groups containing old males (>15 years). One explanation for this is that the older males are more experienced and can keep the group under control with fewer overt shows of aggression [33]. However, the most likely explanation is that colony management strategies are creating a selection bias. With a young male in his first group, we do not know how he will behave towards the females. Those males that become or remain highly aggressive as they age are likely to be removed from breeding, as was the case with the male “Ruger” in Section 3.3/Figure 4. Other males may start with a medium to high level of aggression but show a reduction in aggression as they age or are moved to a different group. These males will probably stay within the colony but would not be mixed with a large group. Of the experienced males, only those with a proven temperament are likely to be introduced to a larger group. This may explain some of the non-linear relationships seen in Figure 6.

One particular recommendation of this study is that young males should be introduced to smaller breeding groups (of up to five females) and should only be moved to larger groups once they have gained some experience and once their behaviour in their first group has been assessed. Rox and colleagues investigated the factors that determine the successful introduction of a male to a group. They found that the younger males were the least likely to be successfully introduced to the groups [20]. Beisner et al. did not find an effect of age on the trauma rate or the success of the introduction. They discuss that this may be due to the skew in their dataset towards younger males, but it may also be because multiple males were introduced to the group, whereas both Rox et al. [20] and this study were looking at single male groups. 

### 4.2. Potential Trade-Off between Injury Rate and Productivity

Some colony management changes that might be aimed at increasing the productivity in a colony may also lead to an increase in injury rate. In particular, the introduction of a new male to a group does increase the pregnancy rate, especially in the first three months after the male is integrated into the group (Figure 5D). However, the introduction of a new male also produces a high injury rate, especially in the introduction phrase (Figure 5A). There are several reasons as to why the injury rate should be particularly high during the introduction phrase. Firstly, the male is a stranger and needs to establish his dominance over the females. Depending on the temperament and experience of the male and of the females within the group this may lead to a spate of injuries as the dominance is established. A second potential issue is that of females fighting among themselves over the new male [19,20,37]. The third potential issue is the disruption caused by separating different members of the group during the introduction phase and confining them to smaller areas of the enclosure. The smaller space means there is less opportunity to escape if there is an aggressive incident and therefore a higher chance of injury. Beisner and colleagues [22] found that trauma rates were lower if the males had continual exposure to the females rather than punctuated exposure [22] and the same group identified introduction enclosures as improving the success of male introductions [23].

Surprisingly we did not see an effect of season on the injury rate. Other groups have found an increase in injuries during the main breeding season [37]. The lack of seasonality with the injury rates may be due to only having one male in the group and so we did not have male competition over females.

### 4.3. Study Limitations and Future Directions

As this was a retrospective analysis of colony health records, we were constrained to only include that information that was recorded reliably and consistently in the health records. Some factors, such as dominance rank, could not be included in the main study as they were not recorded reliably during this period. Other factors, such as the identification of the animals causing the injuries and the time of day the injuries occurred, would only be possible to obtain through constant monitoring. For this purpose we recently installed cameras in the colony to monitor the groups for 24 h each day and allow us to play back aggressive interactions. It was also not possible to obtain a reliable severity score of the injuries from the records. A new injury scoring system was implemented in 2021 and data on this will be available in future.

A final limitation of this study relates to the estimation of conception data. We had to calculate the most likely date of conception based on the date of birth and the average gestation length. Silk [30] showed that there is an effect of maternal age with older females (>15 years) having the shortest gestation period (average of 162 days) and the youngest females (<5 years) the longest (average of 169 days). This means there is likely to be some errors in the actual month of conception compared to our estimate. We have rerun the model with longer gestation periods for the younger animals and shorter gestation periods for the older females and it did not change the outcome of the model.

## 5. Conclusions

As the breeding females were disproportionately likely to be injured, we recommend that any strategies aimed at reducing injuries (for example, use of enrichment and visual barriers) should be focused on these females. The model of injury rate for the breeding females suggests that young breeding males (<7 years) struggle to cope with larger breeding groups (>10 females). We recommend keeping young males in small groups of ~5 breeding females, especially if it is their first breeding group.

## Figures and Tables

**Figure 1 biology-11-00979-f001:**
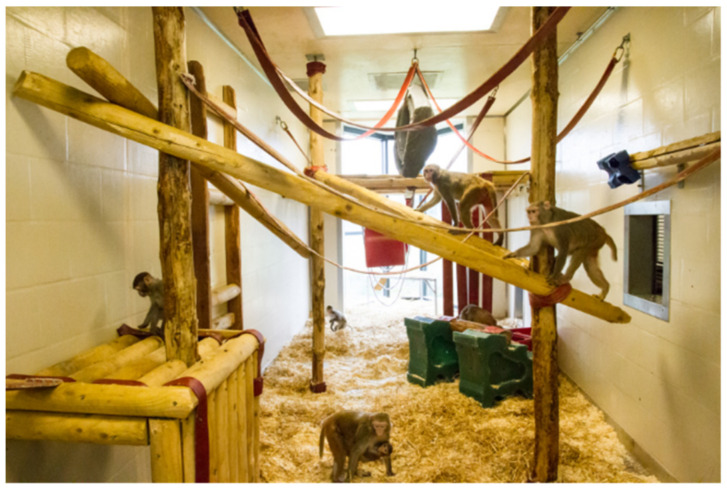
Photograph showing a typical enclosure.

**Figure 2 biology-11-00979-f002:**
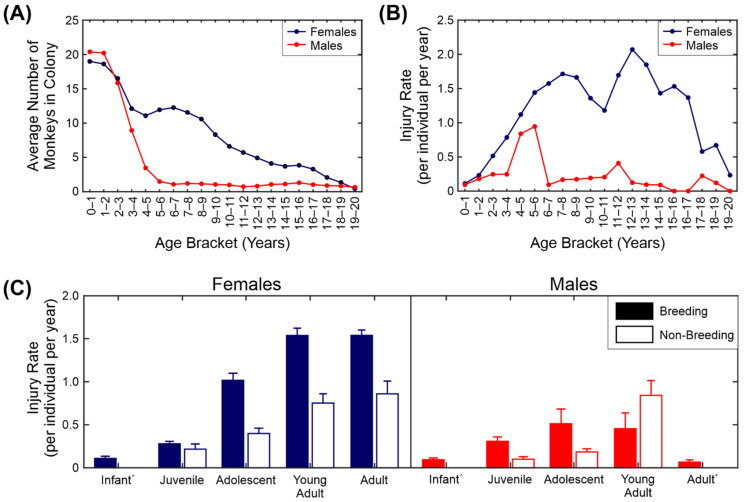
Injury rates across sex, age and group type. (**A**) Distribution of ages and sexes in the colony. (**B**) Average injury rate by age and sex. (**C**) Average injury rate by age category, sex and group type. ^+^ Non-breeding group data not shown for these categories due to insufficient data.

**Figure 3 biology-11-00979-f003:**
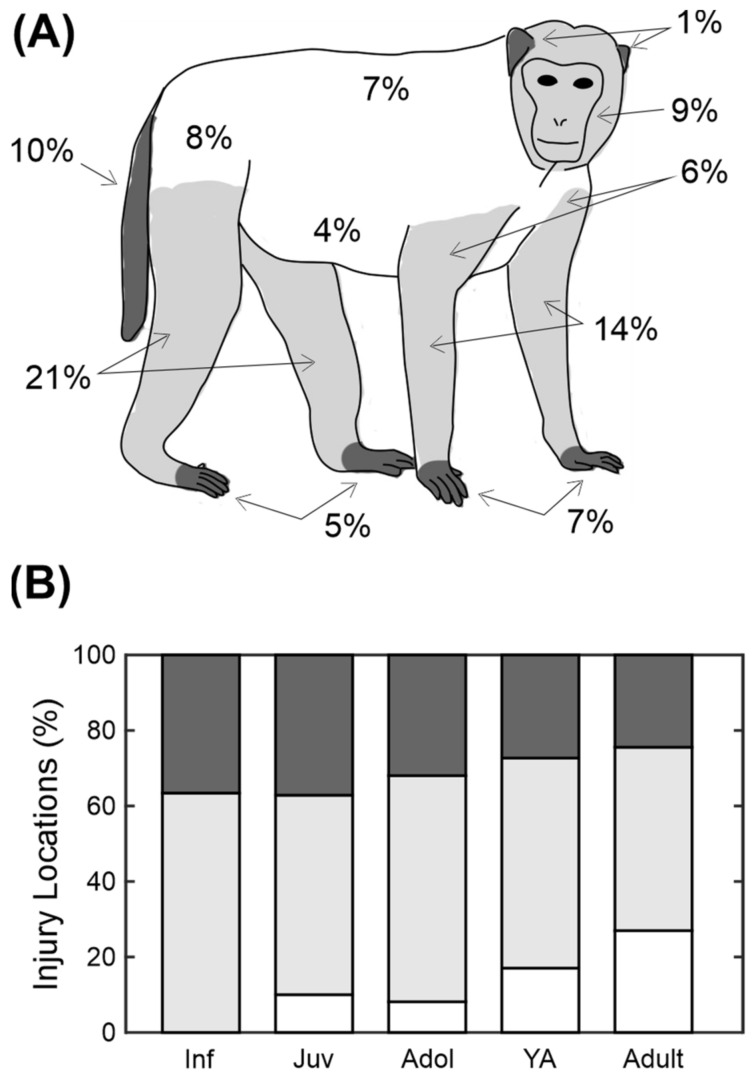
Distribution of injuries across the body. (**A**) Schematic showing the percentage of the injuries that occurred on that body part. (**B**) Distribution of injury location across the different age categories. Shading represents the areas shown in (**A**). Category abbreviations are Inf = infants, Juv = juveniles, Adol = adolescents and YA = young adults.

**Figure 4 biology-11-00979-f004:**
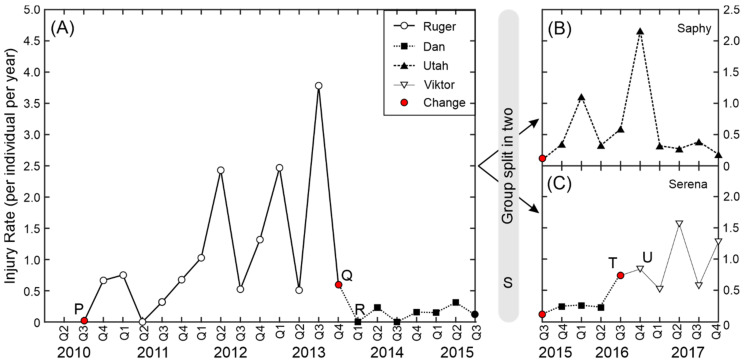
Injury rate for one group between 2010 and 2017 (group split in two in 2015). (**A**) Group before spilt. (**B**) Saphy’s group after split. (**C**) Serena’s group after split. See main text for explanation of letters P–U. Symbols represent the different adult males that were in the group.

**Figure 5 biology-11-00979-f005:**
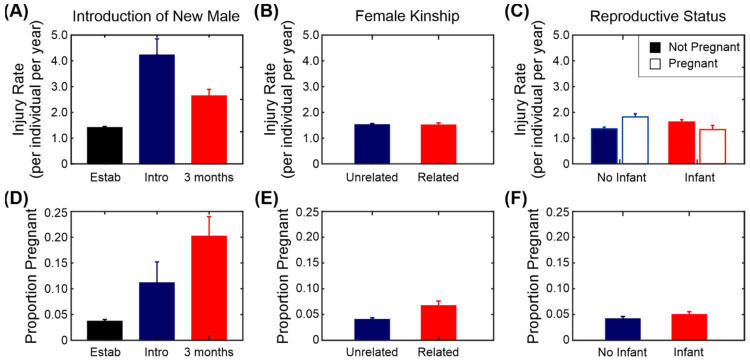
Injury rate and productivity of breeding females. (**A**) Injury rate and introduction of a new breeding male (“Estab” = established group, “Intro” = introduction phase and “3 months” = first 3 months after integration). (**B**) Injury rate and female kinship. (**C**) Injury rate and reproductive status. (**A**–**C**) All results given as mean injury rate ± standard error. (**D**) Productivity (proportion of animals that became pregnant in that month) and the introduction of a new male. (**E**) Productivity and female kinship. (**F**) Productivity and reproductive status. (**D**–**F**) All results given as proportion ± standard error of proportion.

**Figure 6 biology-11-00979-f006:**
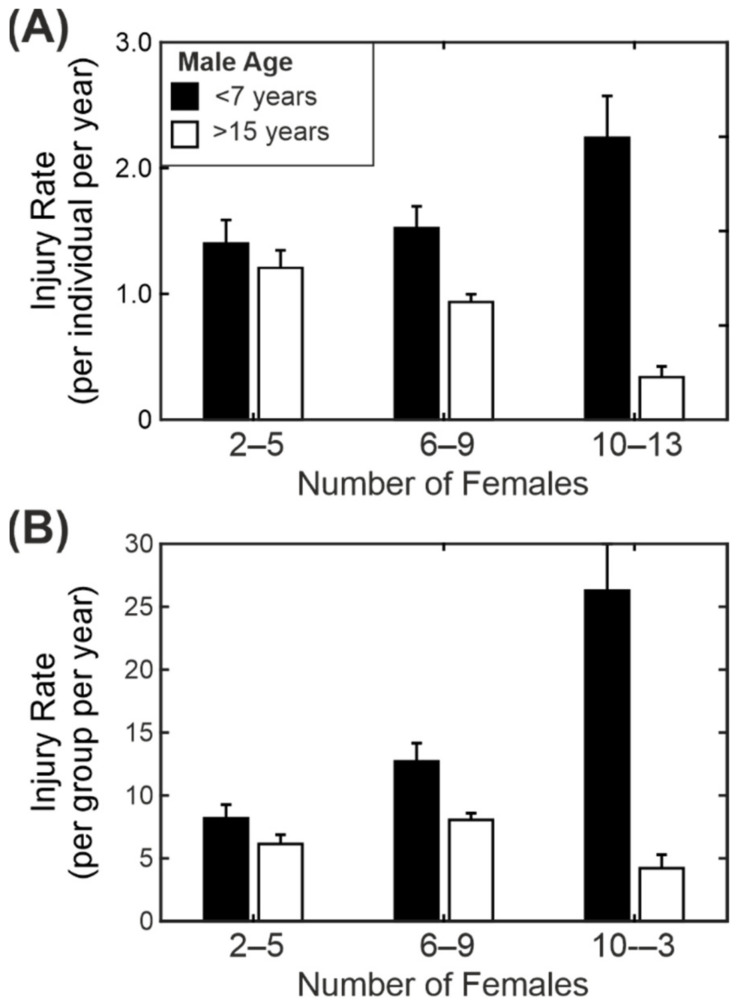
Interaction between the age of the breeding male and the number of breeding females. (**A**) Individual injury rate for different numbers of breeding females. (**B**) Group injury rates for different numbers of breeding females. Results shown as mean injury rate ± standard error for young males (black bars; <7 years age) and old males (white bars; >15 years age).

**Figure 7 biology-11-00979-f007:**
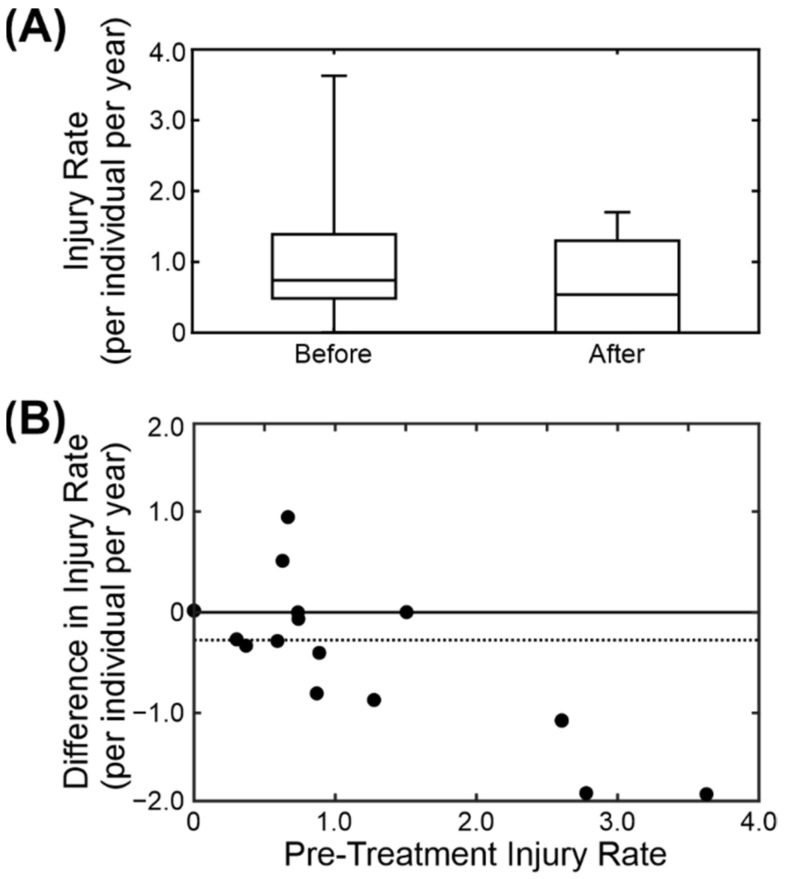
Comparison of injury rate before and after dental treatment on the breeding male. (**A**) Box-and-whisker plot of injury for the 3 months before and the 3 months after the dental treatment on 16 males. (**B**) Scatter plot comparing the pre-treatment injury rate and the difference in injury rate pre- and post-treatment.

**Table 1 biology-11-00979-t001:** Definition of predictor variables.

Predictor Variable	Type	Definition
Sex ^1^	Factor	Whether the animal is “Male” or “Female”
Age ^1^	Continuous	Age of individual animal
Age Class ^1^	Factor	Whether the animal is an “Infant”, “Juvenile”, “Adolescent”, “Young adult” or “Adult”; see Section 2.5 for detailed definition
Group Size ^2^	Discrete	Total number of animals in group
Breeding Group ^2^	Logical	True if animal is in a breeding group; false if not
Oldest Female ^2^	Continuous	Age of the oldest female in the group
UnrelatedFemales ^2^	Logical	False if all the females in the group are related; true if not
New Male ^2^	Factor	“Established”, “Introduction” or “First 3 months”; see Section 3.4 for more details
Pregnancy ^1^	Logical	True if the female is pregnant; false if not
Has Infant ^1^	Logical	True if the female has a living infant of less than 1 year old; false if not
Age of Breeding Male ^2^	Continuous	Age of the breeding male in the group
Number of Females ^2^	Discrete	Number of breeding females in the group
Season ^2^	Factor	Four seasons: December to February (main mating season), March to May, June to August (main birth season) and September to November

^1^ Individual level predictors. ^2^ group level predictors. Grey shaded cells indicate the predictor variables used for the breeding females in results Section 3.4 and Section 3.5.

**Table 2 biology-11-00979-t002:** Summary of the age classes.

Age Class	Age Range	Males		Females		Total	
	(Years)	N ^1^	M ^2^	N ^1^	M ^2^	N ^1^	M ^2^
Infant	0–1.0	232	2549	209	2372	441	4921
Juvenile	1.0–2.5	234	3520	220	3322	454	6842
Adolescent	2.5–4.0	160	1935	172	2394	332	4329
Young Adult	4.0–7.0	60	679	182	4265	242	4944
Adult	7.0+	29	1524	202	8084	231	9608

^1^ N is the number of individual animals in the class. ^2^ M is the number of individual data points in the class. Grey shaded cells indicate the female animals included in the analysis in results Section 3.4 and Section 3.5

**Table 3 biology-11-00979-t003:** Results of the negative binomial mixed-effects model examining the interaction between sex and age class on injury count.

Predictor Variable	Estimate	SE	Z	P	95% Confidence Intervals
Intercept	2.93	0.165	−17.78	< 2 × 10^−16^ ***	−3.26 to −2.61
**Fixed Terms**					
Sex ^1^: Male	−2.79	0.323	−8.65	<2 × 10^−16^ ***	−3.42 to −2.16
Age Class ^2^: Infant	−2.82	0.231	−12.22	<2 × 10^−16^ ***	−3.28 to −2.37
Age Class ^2^: Juvenile	−1.80	0.140	−12.83	<2 × 10^−16^ ***	−2.07 to −1.52
Age Class ^2^: Adolescent	−0.67	0.322	−6.05	1.46 × 10^−9^ ***	−0.89 to −0.45
Age Class ^2^: Young Adult	−0.09	0.074	−1.28	0.2005	−0.24 to 0.05
Group Size	−0.03	0.008	−3.67	0.0002 ***	−0.05 to −0.01
Breeding Group: True	0.72	0.192	3.77	0.0002 ***	0.35 to 1.10
**Interactions**					
Male × Infant ^3^	2.71	0.453	5.96	2.53 × 10^−9^ ***	1.82 to 3.60
Male × Juvenile ^3^	2.75	0.377	7.28	3.25 × 10^−13^ ***	2.00 to 3.48
Male × Adolescent ^3^	1.94	0.397	4.88	1.05 × 10^−6^ ***	1.16 to 2.72
Male × Young Adult ^3^	2.17	0.387	5.61	2.06 × 10^−8^ ***	1.41 to 2.93
**Random terms**	**Variance**	**N**			
Animal ID ^4^	0.369	708			
Unique Group ID ^4^	0.354	98			

^1^ The reference category is female. ^2^ The reference category is adult. ^3^ Interaction between sex and age class. ^4^ Random intercepts were fitted for animal ID and unique group ID. *** Significance *p* < 0.001.

**Table 4 biology-11-00979-t004:** Estimates for interaction between sex and age class (subset of results in Table 3).

Predictor Variable	Female ^1^	Male
Age Class: Infant	−2.82	−2.90
Age Class: Juvenile	−1.80	−1.84
Age Class: Adolescent	−0.67	−1.52
Age Class: Young Adult	−0.09	−0.71
Age Class: Adult ^1^	0	−2.79

^1^ Results shown relative to the adult female class.

**Table 5 biology-11-00979-t005:** Results of the negative binomial mixed-effects model examining the factors affecting injury rate in breeding females.

Predictor Variable	Estimate	SE	Z	P	95% Confidence Intervals
Intercept	−2.44	0.18	−13.24	<2 × 10^−16^ ***	−2.81 to −2.09
**Fixed terms**					
Age	0.06	0.014	4.70	2.67 × 10^−6^ ***	0.03 to 0.09
Oldest Female	−0.02	0.016	−1.00	0.3169	−0.05 to 0.01
Unrelated Females: True	0.03	0.184	0.17	0.8681	−0.33 to −0.39
Season: Mar–May ^1^	−0.04	0.081	−0.51	0.6112	−0.20 to 0.12
Season: Jun–Aug ^1^	−0.09	0.081	−1.07	0.2844	−0.25 to 0.07
Season: Sep–Nov ^1^	0.08	0.078	1.09	0.2737	−0.07 to 0.24
New Male: Introduction ^2^	1.07	0.154	6.88	6.13 × 10^−12^ ***	0.76 to 1.36
New Male: First 3 Months ^2^	0.44	0.110	3.91	9.36 ×10^−5^ ***	0.21 to 0.65
Pregnancy: True	0.10	0.074	1.13	0.2568	−0.06 to 0.23
Has Infant: True	0.02	0.068	0.32	0.7482	−0.11 to 0.16
Age of Breeding Male	−0.07	0.009	−8.07	3.37 × 10^−16^ ***	−0.09 to −0.06
Number of Females	−0.10	0.021	−4.65	0.0001 ***	−0.14 to −0.06
**Interactions**					
Age of Male × No. of Females ^3^	−0.014	0.003	−3.86	0.0001 ***	−0.020 to −0.007
**Random terms**	**Variance**	**N**			
Animal ID ^4^	0.277	224			
Unique Group ID ^4^	0.163	30			

^1^ The reference category is the mating season (December–February). ^2^ The reference category is the established groups. ^3^ Interaction between the age of the breeding male and the number of breeding females. ^4^ Random intercepts were fitted for animal ID and unique group ID. *** Significance *p* < 0.001.

**Table 6 biology-11-00979-t006:** Results of the negative binomial mixed-effects model examining the factors affecting conception rate in breeding females.

Predictor Variable	Estimate	SE	Z	P	95% Confidence Intervals
Intercept	−2.73	0.217	−12.57	<2 × 10^−16^ ***	−3.16 to −2.31
**Fixed terms**					
Age	0.02	0.020	1.12	0.2642	−0.02 to 0.06
Oldest Female	−0.06	0.024	−2.44	0.0146 *	−0.11 to −0.01
Unrelated Females: True	−0.08	0.221	−0.34	0.7310	−0.51 to 0.36
Season: Mar–May ^1^	−0.13	0.130	−1.03	0.3029	−0.39 to 0.12
Season: Jun–Aug ^1^	−1.68	0.201	−8.38	<2 × 10^−16^ ***	−2.07 to −1.29
Season: Sep–Nov ^1^	−1.21	0.158	−7.66	1.89 × 10^−14^ ***	−1.51 to −0.90
New Male: Introduction ^2^	1.11	0.276	3.94	8.31 × 10^−5^ ***	0.55 to 1.63
New Male: First 3 Months ^2^	1.83	0.169	11.13	<2 × 10^−16^ ***	1.55 to 2.21
Has Infant: True	0.03	0.148	0.22	0.8243	−0.26 to 0.32
Age of Breeding Male	−0.01	0.037	−1.78	0.0753	−0.06 to 0.01
Number of Females	−0.01	0.037	−0.28	0.7760	−0.08 to 0.06
**Interactions**					
Age of Male × No. of Females ^3^	−0.009	0.005	−1.66	0.0966	−0.019 to 0.002
**Random terms**	**Variance**	**N**			
Animal ID ^4^	0.070	222			
Unique Group ID ^4^	0.134	30			

^1^ The reference category is the mating season (December–February). ^2^ The reference category is the established groups. ^3^ Interaction between the age of the breeding male and the number of breeding females. ^4^ Random intercepts were fitted for animal ID and unique group ID. * Significance *p* < 0.05, *** Significance *p* < 0.001.

## Data Availability

The data are available in the Appendix A.

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
