# Peer review of "Factors Associated with Injury Rate and Pregnancy Success in Rhesus Macaques"

_biology, 2022, doi:10.3390/biology11070979_

Round 1

Reviewer 1 Report

Dear author,

Thank you for your effort collecting this data and to write this manuscript. The study focus on males, injuries and pregnancy success.

In general, I miss the rank of the involved injured females as rhesus females are known to fight for rank and not only for access to sex (although indirect that is the same) because some families grow in number while others do not so they fight for rank order and not only for breeding succes, months of the conflicts and introduction as rhesus monkeys are seasonal breeders. In which season did you introduce the males? The conclusion is not really a novelty and in my opinion weak. Are there not more new insights obtained from this study?

Minor remarks:

When I read the title, my first impression was: the more injuries, the more babies are born. But that is not the conclusion of your story so I would suggest to adapt the title which is in line with the major finding of this manuscript.

The moment of male introduction can be important as rhesus monkeys have a mating and breeding season. Influence of season? I would expect the most conflicts during the mating season.

Moments of the day in which the in juries occurred? Feeding moment?

Are the conflicts really sex related-introduction related or rank related? Therefore I want to read more about the rank status of the females and not just old or young females! Age says nothing about rank.

Rank of the involved animals? More alfa family injured or families who want to go up in the hierarchy ladder of the group? Alfa females are expected to have access to more sex then lower ranked females so maybe fight more with the male? You focus on age, eg line 365 but I would prefer the rank position

You focus on the age of the males 477-478 and assume that older males are more experienced. But I miss the origine of the males i.e. alfa female origin or low ranked origin as that would maybe influence his attitude and behavior and of course I want to know if they had experience in other groups. Older means only being more experienced when he was in a group before. Maybe the time/month of introduction is more important then you describe (introduction in or outside the mating season) and the experience of the male, the kilo’s he weigh, his experience and not only his age.

I miss the references:

-       Bercovitch, F.B. Reproductive strategies of rhesus macaques. Primates 38, 247–263 (1997). https://doi.org/10.1007/BF02381613

-       Brianne A. Beisner;Caren M. Remillard;Shannon Moss;Caroline E. Long;Kelly L. Bailey;Leigh A. Young;Tracy Meeker;Brenda McCowan;Mollie A. Bloomsmith; (2021). Factors influencing the success of male introductions into groups of female rhesus macaques: Introduction technique, male characteristics and female behavior . American Journal of Primatology, (), –. doi:10.1002/ajp.23314 

-       Bailey, K., Young, L. A., Long, C., Remillard, C., Moss, S., Meeker, T., &Bloomsmith, M. A. (2021). Use of introduction enclosures to integrate multimale cohorts into groups of female rhesus macaques (Macaca mulatta). Journal of the American Association for Laboratory Animal Science, 60, 103–111. https://doi.org/10.30802/ aalas-jaalas-20-000026

-       Rox, A., van Vliet, A. H., Langermans, J. A. M., & Sterck, E. H. M. (2021). A stepwise male introduction procedure to prevent inbreeding in naturalistic macaque breeding groups. Animals: An Open Access Journal from MDPI, 11, 545.

These references should be read and implemented in the manuscript to clarify the novelty of the submitted manuscript.

The conclusion (chapter 5) is quite weak. I really miss the novelty. Sorry.

Reviewer 2 Report

This paper aims at examining the factors associated with injuries and pregnancy in captive colonies of Rhesus macaques. To this aim the authors analysed ten years of medical records  of capive colonies of different compositions.

I read the papers several times, and it looked a bit redundant at the beginnning, but then I realised that the information there was all needed, and logically related to the aim of the study.

This is a very useful paper, and can be of great help for laboratories still housing macaques for scientific research. So, I recommend the publication in "Biology", after very minor editing:

i) The paper present many tables and figures. In some cases there are repetitons of the same data in different format. I would limit these cases, to make the paper more reader-friendly. For example, can the authors choose between Figure 2.A-B and Figure 2.C..? I understand they say the same things. The same goes for Table 3 and Table 4, can one of the two tables be omitted?

ii) I wonder if the authors have any information on the existance of the same kind of data for Macaca fascicularis, another NHP heavily used in biomedical and toxicological research.

Having said thta, the paper deserves smooth publication.

Author Response

This paper aims at examining the factors associated with injuries and pregnancy in captive colonies of Rhesus macaques. To this aim the authors analysed ten years of medical records  of capive colonies of different compositions.

I read the papers several times, and it looked a bit redundant at the beginnning, but then I realised that the information there was all needed, and logically related to the aim of the study.

This is a very useful paper, and can be of great help for laboratories still housing macaques for scientific research. So, I recommend the publication in "Biology", after very minor editing:

Thank you for your comments

i) The paper present many tables and figures. In some cases there are repetitons of the same data in different format. I would limit these cases, to make the paper more reader-friendly. For example, can the authors choose between Figure 2.A-B and Figure 2.C..? I understand they say the same things. The same goes for Table 3 and Table 4, can one of the two tables be omitted?

Figure 2B shows the injury rate across the different ages and sexes and makes it easy to compare males and females at different ages. Figure 2C focusses on the breeding vs non-breeding groups albeit again for different ages and sexes. We considered moving one of these figures into the supplementary material but decided to leave it in the main manuscript.

For Tables 3 and 4 originally we only had table 3 but one of the first readers commented that the interaction term was hard to interpret. We added table 4 to make this easier.

ii) I wonder if the authors have any information on the existance of the same kind of data for Macaca fascicularis, another NHP heavily used in biomedical and toxicological research.

Unfortunately we only have access to rhesus macaques - the majority of similar studies have focussed on rhesus macaques perhaps because they are thought to be more despotic. It would be an interesting comparison though.

Having said thta, the paper deserves smooth publication.

Thank you

Reviewer 3 Report

Review: Factors Associated with Injury Rate and Pregnancy Success in Rhesus Macaques

This research reviews animal records over the past 10 years to ascertain risk factors of injury rate and predictors of pregnancy success in lab-housed rhesus macaques.

This is an interesting and over-due paper that sheds light on a major welfare concern to all captive-housed macaques, not just rhesus and not just those in the lab context. Zoos will also find this information beneficial for shaping husbandry and management.

Figure 3B is not referenced in the main text.

Line 378: does not read correctly.

Line 438: + should be ±

In the method section can the authors please further define pregnancy success or ‘productivity’ – if this was the probability of becoming pregnant how was this determined – the number of copulations, number of cycles? Please discuss further the criteria for determining a monkey was pregnant – the marker of 166 days is the average gestation length – the Silk et al (1993) paper also states that older females had longer pregnancies with heavier infants born. Thus the predictor variable used here of oldest female may be highly related to the output variable – are they statistically independent?  Can the authors estimate the error in calculated month of pregnancy – if maximum pregnancy length is used for groups with relatively old breeding females are the same results achieved? Either more analysis or more evaluation of this as a possible limitation/source of bias is needed in the Discussion. Given the sources of data, evaluation in the Discussion is sufficient. 

Table 1 row ‘Age of Breeding Male’ bottom of the front on the first line (‘age of breeding’) has been cut off – please increase row width.

Overall, a sound and interesting study that makes sense of a large data set in light of the ethology of the species. A worthwhile study and very good quality analysis and presentation of manuscript. 

Author Response

This research reviews animal records over the past 10 years to ascertain risk factors of injury rate and predictors of pregnancy success in lab-housed rhesus macaques.

This is an interesting and over-due paper that sheds light on a major welfare concern to all captive-housed macaques, not just rhesus and not just those in the lab context. Zoos will also find this information beneficial for shaping husbandry and management.

Thank you

Figure 3B is not referenced in the main text.

Fixed thanks

Line 378: does not read correctly.

Edited thanks

Line 438: + should be ±

Corrected thanks

In the method section can the authors please further define pregnancy success or ‘productivity’ – if this was the probability of becoming pregnant how was this determined – the number of copulations, number of cycles? Please discuss further the criteria for determining a monkey was pregnant – the marker of 166 days is the average gestation length – the Silk et al (1993) paper also states that older females had longer pregnancies with heavier infants born. Thus the predictor variable used here of oldest female may be highly related to the output variable – are they statistically independent?  Can the authors estimate the error in calculated month of pregnancy – if maximum pregnancy length is used for groups with relatively old breeding females are the same results achieved? Either more analysis or more evaluation of this as a possible limitation/source of bias is needed in the Discussion. Given the sources of data, evaluation in the Discussion is sufficient. 

Thanks for bringing this to our attention. There seems to be a discrepancy between the abstract of the Silk paper and the main text. The main results and figures suggest that older females have heavier babies but shorter pregnancies (unlike the abstract which said heavier babies and longer pregnancies). They did say age accounted for less than 2% of the variation in gestation length. 

The productivity was calculated as the proportion of females becoming pregnant in that month - unfortunately we do not have data on the number of cycles or copulations in the health records. We reran the model allowing for longer gestation periods in younger females (69 days for females under 8 years) and shorted gestation periods in older females (62 days for females over 12 years). This changed the estimated month of conception for 30 females. It did not affect the outcome of the model apart for some small differences in the estimates. We have added a section to the discussion on limitations (section 4.3; lines 553-560) and discussed this. The oldest female factor refers the age of the oldest female in the group not the age of the individual female (the age of the individual female did not have a significant effect on conception).

Table 1 row ‘Age of Breeding Male’ bottom of the front on the first line (‘age of breeding’) has been cut off – please increase row width.

This has been corrected thanks

Overall, a sound and interesting study that makes sense of a large data set in light of the ethology of the species. A worthwhile study and very good quality analysis and presentation of manuscript. 

Thank you very much

Round 2

Reviewer 1 Report

...